# Microstructures and Mechanical Properties of an AlCoCrNiFe HEA/WC Reinforcing Particle Composite Coating Prepared by Laser Cladding

**DOI:** 10.3390/ma15228020

**Published:** 2022-11-14

**Authors:** Jiang Huang, Zhikai Zhu, Kaiyue Li, Wenqing Shi, Yang Zhao, Minyi He

**Affiliations:** School of Electronics and Information Engineering, Guangdong Ocean University, Zhanjiang 524088, China

**Keywords:** HEA, WC particles, laser cladding, micro-hardness, corrosion resistance

## Abstract

In this study, an AlCoCrFeNi HEA coating with a 10% mass fraction of WC particles was fabricated on the surface of 316L stainless steel by laser cladding technology. WC powders were formed by the partial or total dissolution of the initial WC particles with different sizes in the AlCoCrFeNi HEA coating. Micron WC particles were dispersed in the coating homogeneously, and millimeter WC particles were deposited on the bottom of coating because of their high density. The addition of the WC powers prompted Columnar dendritic and cellular grains, observed in the bottom and top regions of the coating, respectively. Additionally, this led to a higher micro-hardness and better corrosion resistance than that of the pure HEA coating.

## 1. Introduction

High entropy alloys (HEAs) are composed of more than five principal components, which were first reported by Yeh and Cantor et al. in 2004 [1,2]. HEAs have the same properties as traditional single-principal-component alloys, and they also have notable advantages, such as an outstanding thermal stability, fine abrasion, high strength, oxidation resistance and corrosion resistance [3,4,5,6,7]. In recent years, HEA have been applied in many fields, such as aerospace, oil pipelines and automobile shells, because of their unique performance. Nowadays, the methods of induction melting and arc melting are widely used in the preparation of HEAs. However, the high cost of HEAs limits their mass production and extensive use [8]. To date, researchers have found that HEAs can be combined with steels as coatings through laser cladding, which can improve their mechanical properties. Furthermore, this method can also reduce the costs due to the high precision, low dissipation and instantaneous heat-proof capacity of laser cladding [9,10].

AlCoCrFeNi HEA-coated materials have been reported in recent years, and some of them have exhibited excellent properties, such as a high corrosion resistance, wear resistance and hardness [11,12,13]. Moreover, the properties can be further improved through the method of adding ceramic powders to the HEA coating as an additive. For example, Li Yutao et al. synthesized a TiC-reinforced AlCoCrFeNi HEA coating in situ by the laser cladding method and conducted an in-depth study of its wear resistance and hardness. They found that the AlCoCrFeNi-20% TiC coating had the best hardness and wear resistance, and the main wear mechanism of the HEA composite coating was abrasive wear [14]. Xiaofeng Li et al. prepared an AlCoCrFeNi-NbC HEA laser cladding coating and found that NbC particles had a strong pinning effect, which inhibited the grain growth of the HEA coating [15]. By using high-velocity oxygen fuel spraying technology, Junpeng Lv et al. fabricated AlCoCrFeNi-50%WC-10Co HEA composite coatings on 316L stainless steel (316Lss) [16]. At present, there are few investigations on AlCoCrFeNi HEA composite coatings with WC reinforcing particles on 316Lss through laser cladding.

In this paper, we use a 10% mass fraction WC ceramic powder and AlCoCrFeNi HEA mixture as a precursor of laser-clad coatings on the surface of 316Lss. We investigated the micro-structure, phase composition, micro-hardness and corrosion resistance of the compound coatings and discussed the mechanism. This research can provide a reference for the applications of AlCoCrFeNi HEA-WC composite coatings.

## 2. Experimental Methods

Using the XL-F2000W fiber laser processing system (model: XL-F2000 W, manufacturer: Maxphotonics Co., Ltd., Shenzhen, China), the AlCoCrFeNi HEA coating with a 10% mass fraction of the spherical WC reinforcing particles was fabricated on the surface of 316Lss, with a 5% mass fraction uncertainty, which is listed in Table 1. The microscopic morphologies of the spherical AlCoCrFeNi HEA powder and WC particles are given in Figure 1, respectively. The average diameter of the HEA powder is 50–90 μm, while the average diameter of the WC particles is 30–50 μm. Laser cladding was achieved using an XL-F2000W fiber laser with the maximum output power of 2 kW, spot diameter of 3 mm, laser scanning speed of 700 mm/min, defocus amount of +2 mm and preset powder as the powder feeding mode.

The specimen was cut from the deposited coating with a size of 10 mm × 10 mm × 1.5 mm using a wire cutting machine and then polished with SiC abrasive paper. The cross-section was characterized as a microstructure, and the surface was tested for its electrochemical properties. In order to test the phase composition and mechanical properties of the cladded coating, the following testing methods were selected. An optical microscope (model: XJL-302/302BD, manufacturer: Yuexian optical instruument Co., Ltd., Guangzhou, China) and a scanning electron microscope (model: Quanta 250 FEG, manufacturer: FEI company, Hillsborough, OR, USA) coupled with an energy dispersive spectrometer (EDS) were used to analyze the microstructure and chemical composition of the coated specimen. The phase constitutions were detected using an X-ray diffractometer (model: XRD-6100, manufacturer: Shimadzu, Shimane-ken, Japan) at a scanning speed of 6 °/min and a diffraction angle ranging from 20° to 90°. The microhardness of the cladding and substrate was measured using a Vickers hardness tester (model: MHVD-1000AT, manufacturer: Yizong precision instrument Co., Ltd., Shanghai, China) with a load of 200 g for 10 s. The corrosion resistance was measured by electrochemical measurements with potentiodynamic polarization curves using a CHI660E electrochemical workstation. The solution was 3.5% NaCl, and the Tafel evolution curves were obtained in a scanning range of −1.3 V to +0.5 V at room temperature.

## 3. Results and Discussions

### 3.1. Morphology of the AlCoCrFeNi HEA Composite Coatings

Figure 2a shows the scanning method designed for the experiment, while Figure 2b shows the surface appearance of the AlCoCrFeNi HEA composite coatings after the laser cladding processes. The AlCoCrFeNi/WC material was formed on the 316Lss substrate. The cladded coating was flat and smooth, there were no hot cracks or pores.

Figure 3a–c shows the cross-sectional OM of three random parts of the laser-cladded specimen, and Figure 3d is the back-scattered electron image of the specimen. It shows that the cladded coating bound to the 316Lss completely, and there are few fractures. WC particles with diameters of 30–50 μm are concentrated on the bottom of the cladded coating. Few WC particles are seen in the middle and at the top of the coating. The laser cladding schematic diagram is displayed in Figure 3e–g. Before laser cladding, the precursor powders of WC and HEA were mixed for over 2 h in order to ensure the uniformity of the sample. The temperature of the coating increased quickly due to the high laser beam energy during cladding, which caused HEA melting. The addition of the WC powders to the molten metal led to a partial or total dissolution of the initial WC particles with different sizes, which enables them to combine with other metal elements [17,18,19,20]. The larger WC particles may be deposited on the bottom of the coating because of their high density compared to that of HEA and their large scale. However, the smaller particles may be distributed over the whole coating. Thus, the coating is already successfully combined with 316Lss.

Figure 4 shows the cross-sectional microstructure of the laser-cladded AlCoCrFeNi/WC composite coating. Figure 4a shows a section of the laser-cladded coating, in which several WC particles can be seen at the bottom of the coating, with diameters of 30–50 μm, as described above. Figure 4b–d exhibits the high-resolution SEM images of the top, middle and bottom areas, respectively. It can be seen that there are some differences between the three regions. Some cellular grains are generated at the top of the laser-cladded coating, and some cellular grains of a larger size than that of the top grains appear in the middle region, while a directionally solidified columnar dendritic structure was observed in the bottom region. The dark particles in Figure 4 are WC micro-particles. They are caused by the incomplete melting of the WC. Figure 3d shows that the WC particles appear at the bottom of the cladded coating because of their high density. The WC particles in Figure 3 are millimeter-sized, which can be observed using a metallographic microscope. However, the sizes of the WC micro-particles in Figure 4c,d are 2–4 μm. They can be observed using a high-resolution electron microscope. The small size of the WC micro-particles can overcome the limitation of its high density. This enables the WC micro-particles to be evenly distributed in the cladded coating, and the WC particles can only appear at the bottom of the cladded coating.

Laser cladding is influenced by many factors, which makes it a typical non-equilibrium solidification process. According to the constitutional undercooling criterion [21,22], the morphology of an HEA crystal is determined by the ratio of the temperature gradient (G) at the solid–liquid interface and the solidification rate (R). Firstly, with the AlCoCrFeNi/WC composite specimen under the effect of a laser beam, the surface melting of the 316Lss matrix occurs, and the elements in the molten pool are mixed with each other through the effect of the energy flow to form a metallurgical bonding coating, which will be explained by the EDS images in the following figure. Then, the molten pool is solidified through the heat dissipation of the substrate, and the strong chilling effect on the flux causes the G/R value to be very large. Therefore, the nucleation rate of the grain is much faster than the growth rate, and the columnar dendritic structure is formed, as shown in Figure 4d. The existence of WC particles can be ascribed to incomplete melting during laser cladding. With the addition of WC, the micro-structure of the cladded coating changes in a clear manner [12]. There is no large-angle grain boundary in the columnar grains. The microstructure can eliminate the transverse grain boundaries, transfer the longitudinal load and improve the plasticity and creep resistance of the coating. With the development of the solid–liquid interface, G decreases, R increases, and the value of G/R decreases, which causes the cooling rate in the middle region to decrease. The grains have enough time to form in the direction of the heat flow, and large-sized cellular grains grow, as shown in Figure 4c. Finally, the solid–liquid interface moves upward to the top region, the molten pool is far from the 316Lss substrate, the latent heat from the crystallization of the liquid metal can be transferred to both the 316Lss substrate and the surroundings environment, and the value of G/R is smallest, which provides suitable conditions for the generation of small-sized cellular grains, as shown in Figure 4b.

### 3.2. Phase Composition

Figure 5 shows the XRD pattern of the AlCoCrFeNi/WC coating. It can be seen that the diffraction peaks with the highest intensity can be identified as the solid solution, with a BCC phase as the major constituent, while the remaining diffraction peaks can be identified as some metal carbides and WC, which is similar to the findings of Penlin Zhang et al. [23]. In effect, the XRD peak at 32° is ascribed to WC, and the peaks at approximately 43° and 45° can be attributed to Fe_3_W_3_C [20]. Furthermore, the peaks of (Cr,Fe)_7_C_3_ (JCPDS: no. 05-0720) and Fe_2_W (JCPDS: no. 03-0920) are shown in Figure 5.

Figure 6 shows the EDS mappings of the coating with different elements. Figure 6b,c shows that the distributions of Al and Co, two principal components of the HEA, in the cladded coating are higher than that of the substrate, but there are some differences with Cr, Fe and Ni, as shown in Figure 6d–f. The distribution of Cr across the whole surface is the same, while the distribution of Ni in the substrate is smaller than that of the cladded coating. The distribution of Fe in the substrate is much higher than that in the cladded coating. As mentioned above, the substrate used in this experiment was 316Lss, and its chemical composition is described in Table 1. It can be observed that the content of Cr in 316Lss is 17%, which is close to the content of HEA. The content of Ni is 12%, being lower than that of HEA, while the content of Fe is approximately 65%, being much higher than that of HEA. Therefore, the phenomenon of the EDS mapping is depicted in Figure 6d–f. In addition, Figure 6g,h shows the obvious presence of elements of W and C in the cladded coating. This result also proves that the WC micro-particles are dispersed in the coating. 

### 3.3. Mechanical Properties of the Cladded Coating

Next, the dilution rate was calculated. Figure 7 shows the OM image and multi-track laser-cladded diagram, respectively. We simulated the cladded profile depicted in Figure 7a, requiring *H*_1_ to be approximately 1.1 mm and *H*_2_ to be approximately 0.2 mm. We employed the formula of the dilution rate [24]:(1)$$\lambda =\frac{H_{2}}{H_{1}+H_{2}}\times \%=\frac{0.2}{0.2+1.1}\times \%=15.38\%$$

According to the interval standard of the 5–20% dilution rate, the dilution rate of the cladded coating is reasonable.

Figure 7b shows the distribution of the main elements. Figure 7c shows the microhardness distribution curve of the laser-cladded HEA/WC composite coating. It can be seen that the microhardness of the HEA/WC coating differs significantly between the substrate, heat affect zone (HAZ) and coating. In the 316Lss substrate, the microhardness is approximately 190 HV, while the microhardness of HAZ is approximately 293 HV, increased by about 1.5 times compared to that of 316Lss. The microhardness of the HEA/WC coating is approximately 577 HV, increased by about 3 times compared to that of 316Lss, which is highly consistent with the results of other studies [25,26]. Firstly, this can be attributed to the grain size, as shown in Figure 4. Generally, the grain size *d* can be expressed as [27]:(2)$$d=a{\left(R_{c}\right)}^{-n}$$
where *R_c_* is the cooling rate, and *a* and *n* are constants related to the materials. According to the cooling rate theory, the higher the cooling rate is, the smaller the grain size will be. The top region of the cladded coating had a higher cooling rate, and it tended to form refined grains, which led to grain refinement strengthening. The grain refinement strengthening led to a higher hardness of the top region of the cladded coating compared to the other regions [28]. Secondly, the melting point of WC was about 3043 K, which is much higher than those of the other elements in the cladded coating, and some WCs still remained in state of particles during the cladding process. The disordered arrangement of atoms causes serious lattice distortion, which leads to solid solution hardening. Solid solution hardening contributes to the microhardness of the cladded coating [29]. Thirdly, the Al and Co elements improvemed the microhardness of the cladded coating [30].

The potentiodynamic polarization evolution curves of the HEA/WC composite cladded coating and 316Lss substrate in the 3.5 wt% NaCl solution are shown in Figure 7d. The corrosion potential (E_corr_) and the corrosion current density (I_corr_) of the HEA/WC composite cladded coating and 316Lss substrate are shown in Table 2 and were measured by employing the Tafel slope extrapolation method [31]. It can be seen that, compared with the 316Lss substrate (E_corr_ = −0.705 V and I_corr_ = 8.184 × 10^−7^), the HEA/WC composite cladded coating has a higher positive corrosion potential (E_corr_ = −0.633 V) and a lower corrosion current density (I_corr_ = 5.921 × 10^−8^). A higher E_corr_ indicates a lower corrosion thermodynamic tendency, and a lower I_corr_ indicates a lower corrosion rate [32]. Therefore, combining the results of Figure 7d and Table 2, the corrosion resistance of the 316Lss is enhanced by the HEA/WC composite cladded coating. In addition, pitting corrosion appears on the surface of 316Lss, which indicates that the HEA/WC composite cladded coating has a stronger pitting resistance. These methods lead to a better corrosion resistance due to the reduced electron extraction efficiencies and/or parasitic light absorption [33].

## 4. Conclusions

In this study, an AlCoCrFeNi HEA laser-cladded coating with a 10% mass fraction of the spherical WC particles was fabricated on 316Lss substrate by laser cladding. The evolutions along with the morphology, microhardness and corrosion resistance of the cladded coating were examined by OM, SEM, EDS, XRD and HD in detail. The following conclusions are thus presented:(1)The morphology of the HEA composite coating was observed and explained by the differences in the flow and density of matrix in the molten pool and constitutional undercooling criterion.(2)The diffraction peaks with the highest intensity were identified as the solid solution, with a BCC phase as the major constituent, while the remaining diffraction peaks were identified as some metal carbides and WC.(3)The microhardnesses of the HAZ and HEA/WC coating were 1.5 times and 3 times that of the 316Lss substrate, respectively. The main reasons for this are the grain refinement strengthening and solid solution hardening.(4)In the electrochemical experiments, compared with the 316Lss substrate (E_corr_ = −0.705 V and I_corr_ = 8.184 × 10^−7^), the HEA/WC coating had a higher positive corrosion potential (E_corr_ = −0.633 V) and a lower corrosion current density (I_corr_ = 5.921 × 10^−8^), which indicates that the corrosion resistance of the 316Lss was enhanced by the HEA/WC composite cladded coating.

## Figures and Tables

**Figure 1 materials-15-08020-f001:**
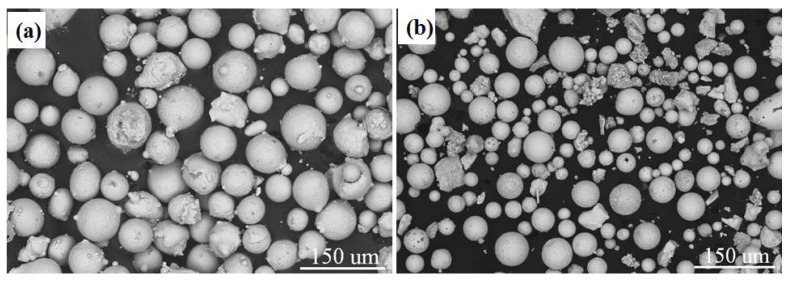
SEM morphology: (**a**) AlCoCrFeNi HEA powders; (**b**) spherical WC powders.

**Figure 2 materials-15-08020-f002:**
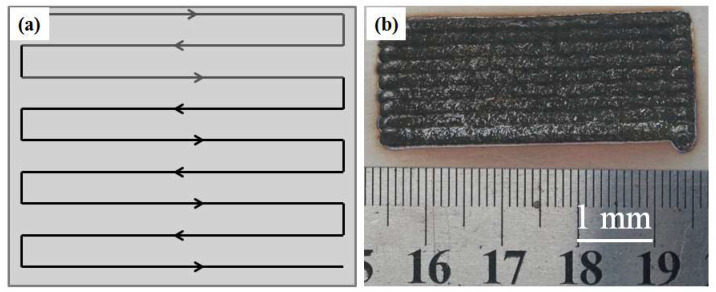
(**a**) The scanning path diagram, and (**b**) the morphology of AlCoCrFeNi HEA coating.

**Figure 3 materials-15-08020-f003:**
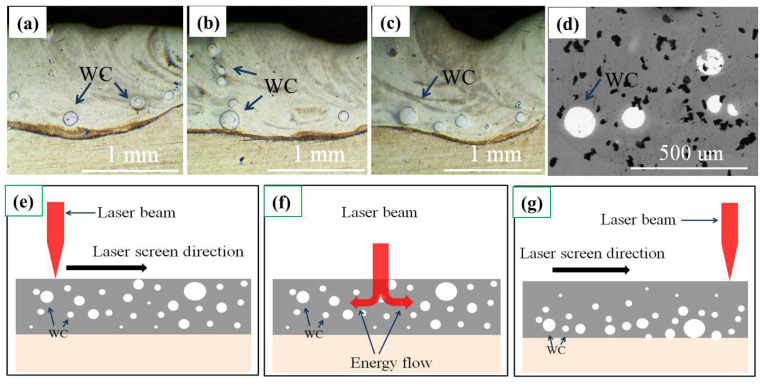
(**a**–**c**) are the OM morphology of the AlCoCrFeNi/WC coating in the different parts; (**d**) is the back-scattered electron image of the AlCoCrFeNi/WC coating; (**e**–**g**) are the schematic diagrams before and after laser cladding.

**Figure 4 materials-15-08020-f004:**
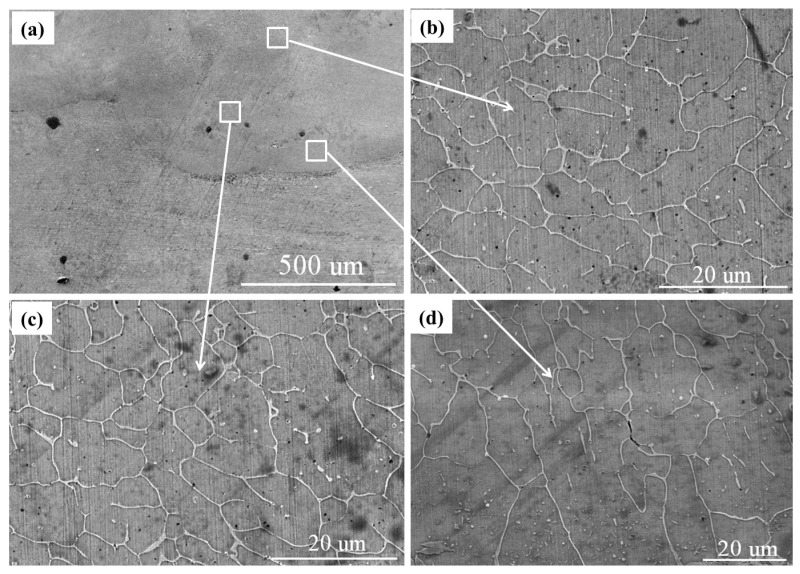
Cross-sectional microstructure of the AlCoCrFeNi/WC coating by laser cladding: (**a**) microstructure of the cladded coating; (**b**–**d**) magnification of the top, middle and bottom regions, respectively.

**Figure 5 materials-15-08020-f005:**
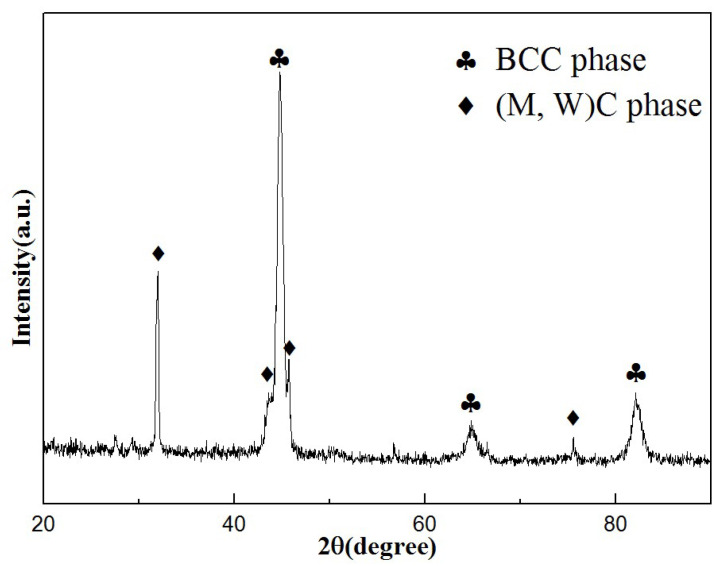
XRD pattern of the laser-cladded AlCoCrFeNi/WC coating.

**Figure 6 materials-15-08020-f006:**
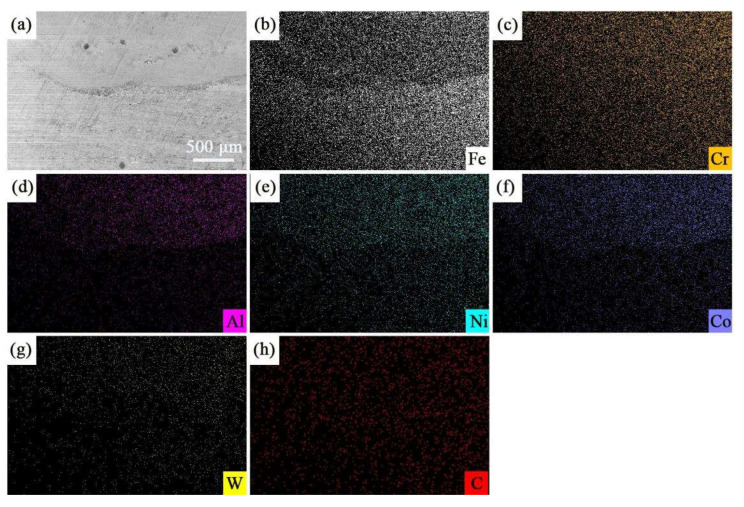
Elemental mapping pictures of the AlCoCrFeNi/WC coating section; (**a**) SEM image of the section and elements of (**b**) Fe, (**c**) Cr, (**d**) Al, (**e**) Ni, (**f**) Co, (**g**) W and (**h**) C.

**Figure 7 materials-15-08020-f007:**
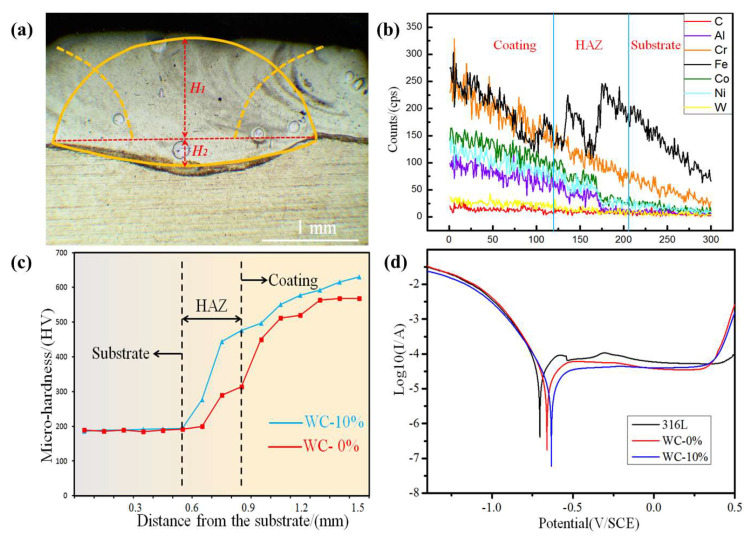
(**a**) The cladding profile of the AlCoCrFeNi/WC coating, (**b**) element distribution curves, (**c**) the microhardness distributions of the AlCoCrFeNi and AlCoCrFeNi/WC coatings, (**d**) the potentiodynamic polarization evolution curves of the coatings.

**Table 1 materials-15-08020-t001:** Chemical composition of 316Lss (mass fraction, %).

Cr	Ni	Mn	Mo	Si	Fe
17	12	2	2.5	1.5	65

**Table 2 materials-15-08020-t002:** The electrochemical parameters of the HEA/WC coatings and 316Lss substrate.

Materials	E_corr_ (V/SCE)	I_corr_ (A/cm^2^)
WC-10%	−0.633	5.921 × 10^−8^
WC-0%	−0.660	1.763 × 10^−7^
316Lss	−0.705	8.184 × 10^−7^

## Data Availability

Not applicable.

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
