# Peer review of "Microstructures and Mechanical Properties of an AlCoCrNiFe HEA/WC Reinforcing Particle Composite Coating Prepared by Laser Cladding"

_materials, 2022, doi:10.3390/ma15228020_

Round 1
Reviewer 1 Report
The article ‘Microstructures and mechanical properties of AlCoCrNiFe HEA/WC reinforcing particles composite coating prepared by laser cladding’ is dedicated to investigate the effect of WC reinforcement to HEA for improving mechanical properties in the laser cladding on 316 SS steel. The manuscript effectively discussed the underlying mechanisms and metallurgical rationale for the acquired results which is its strength and can be deemed suitable as publishable after addressing the following issues:
1. Abstract must include quantitative results acquired through the study.
2. Authors need to explicitly mention the novelty of the present work with respect to the previously published works [16], [22], and [23] which discusses WC addition to HEAs, and laser cladding.
3. The article requires thorough proof reading, it contains numerous English language issues like:
a. L-17-18 is incomplete
b. L-30-31 is incomplete
c. L-33-34 should be single line
d. L-37, it is better to write previous reports rather than existing reports, it seems ambiguous.
e. L-61-62, µ symbol should be used.
f. Wire cutting machine should be replaced by wire-cut EDM machine.
g. L-129 fart?
h. 316 SS
i. L-145 seem
4. What exposure time was used before the electrochemical analysis to obtain stable OCP value?
5. In Figure 3 and Figure 6, ‘Coating/cladding cross-section’ must be included in their captions.
6. Figure 4 contains some cracks and may be inclusions (dark features). Authors fails to discuss these. Point EDS of the inclusions may help.
7. L-135, it is better to designate the regions corresponding to the one mentioned in L-114.
8. The purpose of phase analysis is defeated in XRD, when the authors does not specify the carbide phases present in the coating. Similarly, generalization like some metal carbides as mentioned in conclusion 2 is not acceptable.
9. What technique was employed to obtain the results of figure 7(b), there is no such mention in the manuscript.
10. Not all the plots are mechanical properties in Figure 7. It must include different captions for each part.
11. Nyquist plot may be useful to ascertain the results along with the Tafle plot.
12. Conclusion 1 does not indicate any conclusive results of the study.
Author Response
Reviewer 1:General comment: words that end with ‘ly’ can be deleted (i.e., successfully fabricated). They don’t add value. The meaning of the sentence remains the same without those words. I’m not making this comment for each case. I’m leave it to the author to correct it.
Response:
We are grateful for the recommendation of the reviewers to accept our manuscript. We are sorry about the poor language of our manuscript. We did plenty of work on this paper for a long time. However, English is not our native language, and the repeated addition and removal of sentences and sections led to poor readability and accuracy. We have also involved native English speakers for language corrections. We really hope that the language level has been substantially improved.
The details of changes in abstract are as follows:
- Line 18: replace the word ‘obviously’.
Response: Thank you for the correction. We have already delete “obviously”.
- Line 23: HEA is composed of
Response: Thank you for the correction. We have already add “is” in our sentence.
- Line 27: Define the ‘field of materials’
Response: Thanks for your suggestion. We have defined the “field of materials” as “field of aerospace, oil pipeline, automobile shell, and so on”
- Line 28: replace production for fabrication
Response: Thanks for the professional suggestion. We have changed all “production” to “fabrication”.
- Line 33-34: delete the line break
Response: We accept this suggestion and delete the line break.
- Line 29: The sentence is not clear. Do you mean that: The industrial production is very limited which makes the production costs high?
Response:
We feel sorry for our poor English. The reviewer is right and we replace the sentence as “The industrial production is very limited which makes the production costs high.”
- Line 37: Replace in the existing reports for ‘The fabrication of XX has been reported in the literature”.
Response:
We accept the suggestion and replace the words already.
- Line 56: Experimental processes= experimental methods
Response:
We accept the suggestion and replace the words already.
- Please add a description of the SEM used for this study at the beginning of the paragraph. Right now is under table 1.
Response:
We have added a description of SEM at the beginning of the paragraph as “ Laser cladding was achieved by XL-F2000W fiber laser with the maximum output power of 2kW, the spot diameter is 3mm, the laser scanning speed is 700mm/min, the defocus amount is +2mm, and powder feeding mode is preset powder.”
- Table 1: what’s the % mass fraction uncertainties? Add an *explaining the meaning of ‘Bal.’
Response: The “% mass” means mass fraction, and is described in Table 1 as “Tab.1 Chemical composition of 316Lss (mass fraction, %)”.
- Line 72: remove ‘.And’. The sentence should start with ‘The hardness’.
Response: We accept the suggestion and remove “And”.
- Before line 66. Add a sentence introducing why you are using all of the experimental techniques described below.
Response: Thanks for the warm suggestion. We have added a sentence “In order to test the phase composition and mechanical properties of the cladded coating, the following testing methods are selected”.
- Line 85: why the material has to be melted? And why the scanning path looks that way?
Response: We used imprecise words and mislead our real intention. The materials absorbed a lot of energy and melted quickly during cladding. And the scanning path is designed with fixed program.
- Line 93: Define ‘OM’.
Response: We accept the suggestion and define “OM”.
- Line 122: rephrase it. You should say ‘as we known’
Response: We accept the suggestion and remove“as we known”.
- Figure 5: what the phase at around 42 degrees?
Response: In effect, the XRD peak of "(M, W)C phase" can signify the mixture of MCs (metal carbides) and WC, MCs means FeC, CrC and AlC, which are consistent with the standard card of JCPDS: no. 35-0799, JCPDS: no.51-0997 and JCPDS: no. 35-0783 respectively.
- Line 148: indicate this information inside Figure 5. It would be helpful to compare both studies.
Response: Your suggestion is accepted and we have already indicated the information and compare both studies.
- Line 160: ‘it can be observed’
Response: We accept the suggestion and change to it can be observed.
- Line 163: remove ‘it is not difficult to understand’
Response: We accept the suggestion and remove “it is not difficult to understand”.
- Line 165: replace rich for high. It is obvious but add a short sentence explaining why.
Response: We accept the suggestion and we replaced “rich” for “high” and add a short sentence as “ The distribution of Cr on the whole surface was the same, while the distribution of Ni in the substrate was smaller than that of the cladded coating. ”
- Figure 6: add the scale used. The title of the figure should includes the type of image taken: is it the SEM?
Response: We have use new Fig 6 replaced the old one.
- Figure 7: add short description for each figure (a-d) in the figure title.
Response: Thanks for your suggestion and we have added short description of figures.
- Section 4. Add an introductory paragraph describing the aim of the paragraph.
Response: We have already added an introductory paragraph as the suggestion said.

Reviewer 2 Report
- General comment: words that end with ‘ly’ can be deleted (i.e., successfully fabricated). They don’t add value. The meaning of the sentence remains the same without those words. I’m not making this comment for each case. I’m leave it to the author to correct it.
- Line 18: replace the word ‘obviously’.
- Line 23: HEA is composed of
- Line 27: Define the ‘field of materials’
- Line 28: replace production for fabrication
- Line 33-34: delete the line break
- Line 29: The sentence is not clear. Do you mean that: The industrial production is very limited which makes the production costs high?
- Line 37: Replace in the existing reports for ‘The fabrication of XX has been reported in the literature”.
- Line 56: Experimental processes= experimental methods
- Please add a description of the SEM used for this study at the beginning of the paragraph. Right now is under table 1.
- Table 1: what’s the % mass fraction uncertainties? Add an *explaining the meaning of ‘Bal.’
- Line 72: remove ‘.And’. The sentence should start with ‘The hardness’.
- Before line 66. Add a sentence introducing why you are using all of the experimental techniques described below.
- Line 85: why the material has to be melted? And why the scanning path looks that way?
- Line 93: Define ‘OM’.
- Line 122: rephase it. You should say ‘as we known’
- Figure 5: what the phase at around 42 degrees?
- Line 248: indicate this information inside Figure 5. It would be helpful to compare both studies.
- Line 160: ‘it can be observed’
- Line 163: remove ‘it is not difficult to understand’
- Line 165: replace rich for high. It is obvious but add a short sentence explaining why.
- Figure 6: add the scale used. The title of the figure should includes the type of image taken: is it the SEM?
- Figure 7: add short description for each figure (a-d) in the figure title.
- Section 4. Add an introductory paragraph describing the aim of the paragraph.
Author Response
Reviewer 2:General comment: words that end with ‘ly’ can be deleted (i.e., successfully fabricated). They don’t add value. The meaning of the sentence remains the same without those words. I’m not making this comment for each case. I’m leave it to the author to correct it.
Response:
We are grateful for the recommendation of the reviewers to accept our manuscript. We are sorry about the poor language of our manuscript. We did plenty of work on this paper for a long time. However, English is not our native language, and the repeated addition and removal of sentences and sections led to poor readability and accuracy. We have also involved native English speakers for language corrections. We really hope that the language level has been substantially improved.
The details of changes in abstract are as follows:
- Line 18: replace the word ‘obviously’.
Response: Thank you for the correction. We have already delete “obviously”.
- Line 23: HEA is composed of
Response: Thank you for the correction. We have already add “is” in our sentence.
- Line 27: Define the ‘field of materials’
Response: Thanks for your suggestion. We have defined the “field of materials” as “field of aerospace, oil pipeline, automobile shell, and so on”
- Line 28: replace production for fabrication
Response: Thanks for the professional suggestion. We have changed all “production” to “fabrication”.
- Line 33-34: delete the line break
Response: We accept this suggestion and delete the line break.
- Line 29: The sentence is not clear. Do you mean that: The industrial production is very limited which makes the production costs high?
Response:
We feel sorry for our poor English. The reviewer is right and we replace the sentence as “The industrial production is very limited which makes the production costs high.”
- Line 37: Replace in the existing reports for ‘The fabrication of XX has been reported in the literature”.
Response:
We accept the suggestion and replace the words already.
- Line 56: Experimental processes= experimental methods
Response:
We accept the suggestion and replace the words already.
- Please add a description of the SEM used for this study at the beginning of the paragraph. Right now is under table 1.
Response:
We have added a description of SEM at the beginning of the paragraph as “ Laser cladding was achieved by XL-F2000W fiber laser with the maximum output power of 2kW, the spot diameter is 3mm, the laser scanning speed is 700mm/min, the defocus amount is +2mm, and powder feeding mode is preset powder.”
- Table 1: what’s the % mass fraction uncertainties? Add an *explaining the meaning of ‘Bal.’
Response: The “% mass” means mass fraction, and is described in Table 1 as “Tab.1 Chemical composition of 316Lss (mass fraction, %)”.
- Line 72: remove ‘.And’. The sentence should start with ‘The hardness’.
Response: We accept the suggestion and remove “And”.
- Before line 66. Add a sentence introducing why you are using all of the experimental techniques described below.
Response: Thanks for the warm suggestion. We have added a sentence “In order to test the phase composition and mechanical properties of the cladded coating, the following testing methods are selected”.
- Line 85: why the material has to be melted? And why the scanning path looks that way?
Response: We used imprecise words and mislead our real intention. The materials absorbed a lot of energy and melted quickly during cladding. And the scanning path is designed with fixed program.
- Line 93: Define ‘OM’.
Response: We accept the suggestion and define “OM”.
- Line 122: rephrase it. You should say ‘as we known’
Response: We accept the suggestion and remove“as we known”.
- Figure 5: what the phase at around 42 degrees?
Response: In effect, the XRD peak of "(M, W)C phase" can signify the mixture of MCs (metal carbides) and WC, MCs means FeC, CrC and AlC, which are consistent with the standard card of JCPDS: no. 35-0799, JCPDS: no.51-0997 and JCPDS: no. 35-0783 respectively.
- Line 148: indicate this information inside Figure 5. It would be helpful to compare both studies.
Response: Your suggestion is accepted and we have already indicated the information and compare both studies.
- Line 160: ‘it can be observed’
Response: We accept the suggestion and change to it can be observed.
- Line 163: remove ‘it is not difficult to understand’
Response: We accept the suggestion and remove “it is not difficult to understand”.
- Line 165: replace rich for high. It is obvious but add a short sentence explaining why.
Response: We accept the suggestion and we replaced “rich” for “high” and add a short sentence as “ The distribution of Cr on the whole surface was the same, while the distribution of Ni in the substrate was smaller than that of the cladded coating. ”
- Figure 6: add the scale used. The title of the figure should includes the type of image taken: is it the SEM?
Response: We have use new Fig 6 replaced the old one.
- Figure 7: add short description for each figure (a-d) in the figure title.
Response: Thanks for your suggestion and we have added short description of figures.
- Section 4. Add an introductory paragraph describing the aim of the paragraph.
Response: We have already added an introductory paragraph as the suggestion said.

Reviewer 3 Report
I. The English language should be corrected and the terminology should be checked throughout the text. In many cases, a language error has actually a terminological meaning and hence distorts the physical meaning of the text. Some examples together with the reviewer's comments on the subject matter of the paper are presented below.
Abstract: "... the microhardness of the coating was obviously." Obviously what?
"...reinforcement of WC particles also improved the corrosion resistance of the coating..." Reinforcement of particles themselves or reinforcement of the coating by particles?
P.1, line 31: "... can the performance be enhanced." Check the English grammar.
P.1, lines 33-34: "... the segregation of compositions can be prevented..." Term "segregation" can refer to components but not to composition.
P.2, line 49 and throughout the text: "316Lss". The steel grade is "316L" but not "316Lss". If the authors mean "stainless steel" under "ss", then this abbreviation is supposed to be decoded when it appears for the first time.
P2. When formulating he research objective on p.2, line 53, the authors claim: "The research results can provide reference for the application of AlCoCrFeNi HEA." This is misleading. As seen from the text, the paper deals with a HEA+WC coating, but not with the application of the HEA itself.
P.2, lines 62-64: "XL-F2000W fiber processing machine ... were applied the prepare the coatings." How many machines were applied? And why did the authors use "fiber processing machine" whereas they prepared the coating from powders rather than from fibers? This issue should be explained in the text. And, since it is the "fiber processing machine", then how the powders were fed?
P.2, line 66: "coating was cut with a size of 10mm x 10mm x 1.5mm..." What it a size of the coating or just the size of specimens cut out from the deposited coating?
P.2, lines 68-69: "Quanta 250 FEG, America..." Which particular America is meant?
P.2, lines 73-74: "...with a measurement interval of 200 g...". What do the authors mean under "measurement interval"? Did they vary the load with this "interval", i.e. increment? Or do they mean anything else under this term?
P.2, line 74: "The corrosion resistances..." To the best of the reviewer's knowledge, whose native language is not English, term "resistance" is used as plural only in Electrical Engineering.
P.2, lines 75-76: "by electrochemical measurement (CHI660E, China) ... in electrochemical workstation..." Is "CHI660E, China" a standard for measurements? Or it refers to the workstation?
P.2, line 77: "Tafel evolution curves were given in a scanning range..." How curves can be "given"? Why not obtained? Or do the authors mean the preset parameters for scanning?
P.3, line 85: "...material was sufficiently melted..." What means "sufficiently"? In engineering sciences, we are supposed to be more precise in definitions.
P.3, lines 86-87: "each single-track passes was almost parallel..." Check the English grammar.
P.3, Caption to Fig.3 (lines 91-92): Why do the authors use the past tense ("were" instead of "are")?
"...the simulation diagram..." In the subsequent text, there isn't a single word about "simulation" of the process, the simulation method, etc. So why do the authors use this term to describe Fig3, (e)-(g)?
P.4, lines 98-99: "Before laser cladding, the coating was almost uniform because the composite powders were fully mixed for 2 hours." The authors do not present the microstructure of the coating "before laser cladding", so, strictly speaking, 2 h mixing is not the proof of the uniformity.
P.4, 1st paragraph from the top. To decribe the structure evolution, the authors cite papers [17-20] on modeling the convection in the melt pool, which originates due to the Marangoni, or Gibbs–Marangoni effect, which, in its physical meaning, is caused by a change in the surface tension of the melt because of the temperature gradient. But, as they state later (lines 107-108), "Since the density of WC particles is much higher ... WC particles are deposited at the bottom of the molten pool...". Actually, the main reason must by the density, and it will work even without any convection. So why do the authors mix two different physical effects to explain sedimentation of particles due to the difference in density?
Later (P.4, lines 127-128), it is stated that the Marangoni convection causes mixing of elements in the melt pool, which is physically correct in this case.
The reviewer should also note that if the Marangoni convection is strong, it may impede the deposition of heavier WC particles to the bottom of the pool. But in this case, physical estimates are necessary.
P.4, line 113 and other places: "cladding coating..." Is it a "cladding coating" or, in fact, a cladded coating?
P.4, line 130: "heat dissipation of the 316Lss substrate..." Why "of the substrate" and not dissipation to the substrate?
P.4, line 133: "There was no large-angle grain boundary in columnar grains..." The authors did not examine the structure of the grain boundaries, nor the misorientation of the grains; the latter can be easily studied using the EBSD method. Besides, in Fig.4(d) it is not clearly seen that the grains are really "columnar dendritic". So, this statement is merely speculative. The authors are supposed to present more convincing microstructures, maybe at higher magnifications, and to prove the absence of large-angle grain boundaries. Or not to speak about the latter. Besides, the authors did not explain what are the dark spots in Fig.4. Are they the WC particles, same as in Fig.3(d)? If so, then it is seen that their number is larger in the middle (Fig.4(c)) then in the bottom (Fig.4(d)) of the coating, which contradicts the author's concept presented in Fig.3(g).
P.5, lines 131-132: "the nucleation rate of the grain is much faster than the growth rate, and the columnar dendritic structure shown was formed..." From the crystallization theory it is known that dendrites grow in the direction opposite to the heat flow, and here the growth rate must be higher than the nucleation rate - otherwise the structure would be equiaxial.
P.5, lines 140-141: "the latent heat of crystallization ... can be derived from both ... substrate and the surroundings environment..." The text is confusing and contradicts the crystallization theory. First, why the heat or crystalization is "derived"? It's a mathematical term. Second, the heat of crystallization releases in any place of the melt where crystallization, i.e. phase transition, occurs.
P.5, line 161 and later in the text: "HEA alloy..." This is tautology because HEA means "high entropy alloy".
P.5, Fig.5: what is "(M, W)C phase" depicted in this figure? If it is a complex carbide, then which particular metal M (or metals) dissolves in WC? In the text (line 148) the authors speak only about WC and "some metal carbide", which actually was not identified whereas the carbides of the carbide-forming metals that are present in the HEA (Cr and Fe) are well known.
P.5. When describing the EDS results (Fig.6), the authors did not mention Fig.6(a) and did not indicate the position of the substrate/coating interface in all the figures. Also, they did not note what indicates high concentration of an element: brighter or darker color? If it is brighter, then it is seen that for W there are more bright points near the top of Fig.6(g) than in the middle, so any reader can conclude that there are more WC particles, which are heavier than the AlCoCrFeNi alloy, in the upper part of the coating than close to the substrate/coating interface. But then all the authors' consideration about the distribution of the WC particles (see Fig.3 and the comments above) appear incorrect. And what was the accuracy of EDS for C? It is known than for light elements it is very low.
And what is "cladding surface" in caption to Fig.6? Is it the cladded surface or anything else?
Fig.7(b): the distribution of W is seen to be uniform over the coating thickness. Once again, this shows that the authors' considerations about the distribution of WC particles across the coating (Fig.3 and in many places of the text) appear to contradict to their own experimental results.
Caption to Fig.7, "mechanical properties of the HEA/WC coating", is misleading: cladding profile (a), element distribution curves (b) and polarization curves (d) have nothing to do with "mechanical properties".
P.6, lines 179-180: According to the interval standard of 5-20% dilution rate, the upper dilution rate was reasonable." The phrase is confusing, the authors probably wanted to say that the obtained value is below the upper limit.
P.7, lines 186-187: "... hard WC particles penetrated into the interface with Marangoni convection, the microhardness of this region increased gradually." First, the WC did not penetrate "into" the interface but are supposed to concentrate near it. Second, as I stated above, in the paper there are no evidences that the fraction of WC particles at the interface is higher than in the central part of the coating. Third, the Vickers hardness of WC is around 2600. If the WC particles really deposited at the interface, the microhardenss near the interface would be substantially higher than away from it. But in Fig.7(c) the situation is the opposite. So, once again, the authors' concept expressed in Fig.3 and a large part of the text concerning WC particles appears misleading.
Conclusion (1): "The morphology ... was observed and explained by Marangoni effect and constitutional undercooling criterion." First, as shown above, the authors' explanation including the Marangoni effect is misleading. Second, the constitutional undercooling criterion is not used in this work.
Conclusion (2): "...the remaining diffraction peaks are identified as some metal carbide and WC." In describing the XDR results presented in Fig.5, the authors did not identify which particular "metal carbide" was it, but they mentioned complex carbide (M,W)C (see comments to Fig.5)
Conclusion (3): "The main reasons are grain refinement strengthening and solid-solution hardening." In the text, the authors mentioned the presence of undissolved WC particles (p.7, line 198), but here they do not describe their contribution to hardening. Then why there was so many reasoning in the text about the role of WC particles?
The reviewer's conclusion: from the above it is clearly seen that, despite a large work done by the authors, their explanations/consideration/reasoning are misleading. Thus, the level of the paper is lower than is passable for a peer-reviewed journal with a high impact factor (as high as 3.748).
Hereby I recommend to reject the paper.
Author Response
Reviewer 3:
- The English language should be corrected and the terminology should be checked throughout the text. In many cases, a language error has actually a terminological meaning and hence distorts the physical meaning of the text. Some examples together with the reviewer's comments on the subject matter of the paper are presented below.
Response:
Thank you for your professional comments. As English is not our mother tongue, it is not so easy to change the English of our manuscript properly. We agree with the suggestion completely, also, we have revised the manuscript in an attempt to increase clarity and minimize unfounded speculation. Following the specific comments, we have undertaken the following modification:
Abstract: "... the microhardness of the coating was obviously." Obviously what?
Response: We regret that we made a obvious mistake, the sentence has been modified and is now: Under the action of the grain refinement strengthening and solid-solution hardening, the microhardness of the coating was enhanced.
"...reinforcement of WC particles also improved the corrosion resistance of the coating..." Reinforcement of particles themselves or reinforcement of the coating by particles?
Response:
The Tafel slope shows that the addition of WC can reduce the corrosion thermodynamic tendency and improve the positive corrosion potential effectively. That may caused by the production of metal carbides during laser cladding with WC. As we known, metal carbides can slow down the speed of electrochemical corrosion. The details is indicated as: “these methods lead to better corrosion resistance due to reduced electron extraction efficiencies and/or parasitic light absorption[33].”
P.1, line 31: "... can the performance be enhanced." Check the English grammar.
Response:
We have revised the “However, only by improving the composition and structure of the surface can the performance be enhanced” to “ However, mechanical properties can be enhanced by improving the composition and structure of the surface.” in paragraph one of introduction.
P.1, lines 33-34: "... the segregation of compositions can be prevented..." Term "segregation" can refer to components but not to composition.
Response:
We have revised the “the segregation of compositions can be prevented” to “ the segregation of components can be prevented” in paragraph one of introduction.
P.2, line 49 and throughout the text: "316Lss". The steel grade is "316L" but not "316Lss". If the authors mean "stainless steel" under "ss", then this abbreviation is supposed to be decoded when it appears for the first time.
Response:
Thanks for your suggestion, we have revised the “By using high-velocity oxygen fuel spraying technology, Junpeng Lv et al. [16] successfully fabricated the AlCoCrFeNi-50%WC -10Co HEA composite coatings on 316Lss” to By using high-velocity oxygen fuel spraying technology, Junpeng Lv et al. fabricated the AlCoCrFeNi-50%WC -10Co HEA composite coatings on 316L stainless steel (316Lss) [16] ” in paragraph twoof introduction.
P2. When formulating he research objective on p.2, line 53, the authors claim: "The research results can provide reference for the application of AlCoCrFeNi HEA." This is misleading. As seen from the text, the paper deals with a HEA+WC coating, but not with the application of the HEA itself.
Response:
Our original intention is to explore a method that can improve the corrosion resistance of 316L stainless steel. In order to make the experimental data more rigorous, we have revised the “The research results can provide reference for the application of AlCoCrFeNi HEA” to “The research results can provide reference for the application of AlCoCrFeNi HEA-WC composite coating” in last paragraph of introduction.
P.2, lines 62-64: "XL-F2000W fiber processing machine ... were applied the prepare the coatings." How many machines were applied? And why did the authors use "fiber processing machine" whereas they prepared the coating from powders rather than from fibers? This issue should be explained in the text. And, since it is the "fiber processing machine", then how the powders were fed?
Response:
We are sorry for the less rigorous words, we have revised the “The XL-F2000W fiber processing machine with the maximum output power of 2kw, the spot diameter of 3mm, the laser scanning speed of 700mm/min, and the defocus amount is +2mm were applied the prepare the coatings.” to “Laser cladding was achieved by XL-F2000W fiber laser with the maximum output power of 2kW, the spot diameter is 3mm, the laser scanning speed is 700mm/min, the defocus amount is +2mm, and powder feeding mode is preset powder.” in experimental methods.
P.2, line 66: "coating was cut with a size of 10mm x 10mm x 1.5mm..." What it a size of the coating or just the size of specimens cut out from the deposited coating?
Response:
In order to make our statement more rigorous, we have revised the “The laser cladding coating was cut with a size of 10mm×10mm×1.5mm by wire cutting machine” to “The specimen was cut out from the deposited coating with a size of 10mm×10mm×1.5mm by wire cutting machine, and then polished with SiC abrasive papers. The cross-section was characterized as microstructure, and the surface was tested as electrochemical properties.” in experimental methods.
P.2, lines 68-69: "Quanta 250 FEG, America..." Which particular America is meant?
Response:
we have revised the “SEM, FEI, Quanta 250 FEG, America” to “SEM, FEI, Quanta 250 FEG, United States” in experimental methods.
P.2, lines 73-74: "...with a measurement interval of 200 g...". What do the authors mean under "measurement interval"? Did they vary the load with this "interval", i.e. increment? Or do they mean anything else under this term?
Response:
In order to make our statement more rigorous, we have revised the “with a measurement interval of 200 g and a load retention time of 10 s.” to “ with a load of 200 g for 10 s” in experimental methods.
P.2, line 74: "The corrosion resistances..." To the best of the reviewer's knowledge, whose native language is not English, term "resistance" is used as plural only in Electrical Engineering.
Response:
In order to make our statement more rigorous, we have revised the “The corrosion resistances were measured by electrochemical measurement (CHI660E, China) with potentiodynamic polarization curves in electrochemical workstation” to “ The corrosion resistance was measured by electrochemical measurement with potentiodynamic polarization curves in CHI660E electrochemical workstation” in experimental methods.
P.2, lines 75-76: "by electrochemical measurement (CHI660E, China) ... in electrochemical workstation..." Is "CHI660E, China" a standard for measurements? Or it refers to the workstation?
Response:
We have revised the “The corrosion resistances were measured by electrochemical measurement (CHI660E, China) with potentiodynamic polarization curves in electrochemical workstation” to “ The corrosion resistance was measured by electrochemical measurement with potentiodynamic polarization curves in CHI660E electrochemical workstation” in experimental methods.
P.2, line 77: "Tafel evolution curves were given in a scanning range..." How curves can be "given"? Why not obtained? Or do the authors mean the preset parameters for scanning?
Response:
The question may have been arisen because of a misunderstanding. We have revised the “Tafel evolution curves were given in a scanning range of -1.3 V to +0.5 V at room temperature” to “ Tafel evolution curves were obtained in a scanning range of -1.3 V to +0.5 V at room temperature.” in experimental methods.
P.3, line 85: "...material was sufficiently melted..." What means "sufficiently"? In engineering sciences, we are supposed to be more precise in definitions.
Response:
In order to make our statement more rigorous, we have revised the “The AlCo- CrFeNi/WC material was sufficiently melted and a dense coating was formed on the 316Lss substrate.” to “ The AlCoCrFeNi/WC material was formed on the 316Lss substrate” in results and discussions.
P.3, lines 86-87: "each single-track passes was almost parallel..." Check the English grammar.
Response:
We have revised the “each single-track passes was almost parallel to each other” to “ The cladded coating is flat and smooth, there are no hot cracks and pores.”
P.3, Caption to Fig.3 (lines 91-92): Why do the authors use the past tense ("were" instead of "are")?
Response:
We have revised the “ Fig. 3 (a)-(c) were the OM morphology of the coating in different parts; (d) was the back scattered electron image of the coating; (e) and (f) were the simulation diagram before and after laser cladding.” to “ Fig. 3 (a)-(c) are the OM morphology of the coating in different parts; (d) is the back scattered electron image of the coating; (e) and (g) are the schematic explanation before and after laser cladding.”
"...the simulation diagram..." In the subsequent text, there isn't a single word about "simulation" of the process, the simulation method, etc. So why do the authors use this term to describe Fig3, (e)-(g)?
Response:
The question may have been arisen because of a misunderstanding, Fig 3, (e)-(g) are the schematic explanation of laser cladding, not the “simulation diagram”. We have changed the annotation of Fig 3.
P.4, lines 98-99: "Before laser cladding, the coating was almost uniform because the composite powders were fully mixed for 2 hours." The authors do not present the microstructure of the coating "before laser cladding", so, strictly speaking, 2 h mixing is not the proof of the uniformity.
Response:
Thanks for the reviewer’s kind reminder, we have corrected the mistake. And we delete “fully”.
P.4, 1st paragraph from the top. To describe the structure evolution, the authors cite papers [17-20] on modeling the convection in the melt pool, which originates due to the Marangoni, or Gibbs–Marangoni effect, which, in its physical meaning, is caused by a change in the surface tension of the melt because of the temperature gradient. But, as they state later (lines 107-108), "Since the density of WC particles is much higher ... WC particles are deposited at the bottom of the molten pool...". Actually, the main reason must by the density, and it will work even without any convection. So why do the authors mix two different physical effects to explain sedimentation of particles due to the difference in density?
Response:
The reviewer is correct in noticing this point of view. We have changed the “Driven by the laser beam, the liquid AlCoCrFeNi HEA/WC composite powders moves from top to bottom, forming Marangoni convection, as shown inFig. 3 (f). Since the density of WC particles is much higher than that of AlCoCrFeNi HEA powders, a large number of WC particles are deposited at the bottom of the molten pool.” to “ Driven by the laser beam, the liquid AlCoCrFeNi HEA/WC composite powders moves from top to bottom, forming energy flow, as shown in Fig. 3 (f). Since the density of WC particles is much higher than that of AlCoCrFeNi HEA powders, a large number of WC particles are deposited at the bottom of the molten pool”.
Later (P.4, lines 127-128), it is stated that the Marangoni convection causes mixing of elements in the melt pool, which is physically correct in this case.
Response:
In order to make sure this logic is more clear and complete, we have changed this sentence from “the elements in the molten pool were mixed with each other by the effect of Marangoni convection to form metallurgical bonding coating” to “ the elements in the molten pool are mixed with each other by the effect of energy flow to form metallurgical bonding coating” .
The reviewer should also note that if the Marangoni convection is strong, it may impede the deposition of heavier WC particles to the bottom of the pool. But in this case, physical estimates are necessary.
Response:
Thanks for reviewer’s detailed reminder. In order to make sure this logic is more clear and complete, we change this sentence from “, the elements in the molten pool were mixed with each other by the effect of Marangoni convection to form metallurgical bonding coating” to “the elements in the molten pool are mixed with each other by the effect of energy flow to form metallurgical bonding coating”
P.4, line 113 and other places: "cladding coating..." Is it a "cladding coating" or, in fact, a cladded coating?
Response:
Thanks for reviewer’s detailed reminder. We accept this suggestion and change all "cladding coating" to “cladded coating”.
P.4, line 130: "heat dissipation of the 316Lss substrate..." Why "of the substrate" and not dissipation to the substrate?
Response:
We have checked this sentence and modified the ambiguous part as: Then, the molten pool was solidified by heat dissipation of substrate,and the strong chilling effect on the flux makes the G/R value very large.
P.4, line 133: "There was no large-angle grain boundary in columnar grains..." The authors did not examine the structure of the grain boundaries, nor the misorientation of the grains; the latter can be easily studied using the EBSD method. Besides, in Fig.4(d) it is not clearly seen that the grains are really "columnar dendritic". So, this statement is merely speculative. The authors are supposed to present more convincing microstructures, maybe at higher magnifications, and to prove the absence of large-angle grain boundaries. Or not to speak about the latter. Besides, the authors did not explain what are the dark spots in Fig.4. Are they the WC particles, same as in Fig.3(d)? If so, then it is seen that their number is larger in the middle (Fig.4(c)) then in the bottom (Fig.4(d)) of the coating, which contradicts the author's concept presented in Fig.3(g).
Response:
The reviewer is correct in noticing this point of view. Limited by conditions, there is no chance to perform the EBSD test. So in our manuscript, we propose a possibility as “There was no large-angle grain boundary in columnar grains. The microstructure may eliminates the transverse grain boundaries, transfers the longitudinal load and improves the plasticity and creep resistance of the coating. With the advance of the solid-liquid interface,G decreases, R increases, and the value of G/R decreases, which makes the cooling rate in the middle region decrease”.
P.5, lines 131-132: "the nucleation rate of the grain is much faster than the growth rate, and the columnar dendritic structure shown was formed..." From the crystallization theory it is known that dendrites grow in the direction opposite to the heat flow, and here the growth rate must be higher than the nucleation rate - otherwise the structure would be equiaxial.
Response:
Thanks for review’s professional suggestion. We accept the suggestion and change the sentence of “Therefore, the nucleation rate of the grain is much faster than the growth rate, and the columnar dendritic structure shown was formed” to “Therefore, the nucleation rate of the grain is much faster than the growth rate, and the columnar dendritic structure shown was formed in Fig. 4 (d)”
P.5, lines 140-141: "the latent heat of crystallization ... can be derived from both ... substrate and the surroundings environment..." The text is confusing and contradicts the crystallization theory. First, why the heat or crystalization is "derived"? It's a mathematical term. Second, the heat of crystallization releases in any place of the melt where crystallization, i.e. phase transition, occurs.
Response:
Thanks for review’s professional suggestion. we have changed this sentence from “,the latent heat of crystallization of the liquid metal can be derived from both the 316Lss substrate and the surroundings environment” to “ the molten pool is far away from the 316Lss substrate, the latent heat of crystallization of the liquid metal can be transferred to both the 316Lss substrate and the surroundings environment, and the value of G/R is smallest” .
P.5, line 161 and later in the text: "HEA alloy..." This is tautology because HEA means "high entropy alloy".
Response:
We have changed this sentence from “which was closed to the content of HEA alloy” to “ which was closed to the content of HEA ” .
P.5, Fig.5: what is "(M, W)C phase" depicted in this figure? If it is a complex carbide, then which particular metal M (or metals) dissolves in WC? In the text (line 148) the authors speak only about WC and "some metal carbide", which actually was not identified whereas the carbides of the carbide-forming metals that are present in the HEA (Cr and Fe) are well known.
Response:
In effect, the XRD peak of "(M, W)C phase" can signify the mixture of MCs (metal carbides) and WC, MCs means FeC, CrC and AlC, which are consistent with the standard card of JCPDS: no. 35-0799, JCPDS: no.51-0997 and JCPDS: no. 35-0783 respectively.
P.5. When describing the EDS results (Fig.6), the authors did not mention Fig.6(a) and did not indicate the position of the substrate/coating interface in all the figures. Also, they did not note what indicates high concentration of an element: brighter or darker color? If it is brighter, then it is seen that for W there are more bright points near the top of Fig.6(g) than in the middle, so any reader can conclude that there are more WC particles, which are heavier than the AlCoCrFeNi alloy, in the upper part of the coating than close to the substrate/coating interface. But then all the authors' consideration about the distribution of the WC particles (see Fig.3 and the comments above) appear incorrect. And what was the accuracy of EDS for C? It is known than for light elements it is very low.
Response: Thanks for your warm suggestion, we have changed Fig 6 instead of the old one.
And what is "cladding surface" in caption to Fig.6? Is it the cladded surface or anything else?
Response: In Fig.6, the cladding surface means the cross section of sample. The details are shown in experimental methods as “The cross-section was characterized as microstructure, and the surface was tested as electrochemical properties”.
Fig.7(b): the distribution of W is seen to be uniform over the coating thickness. Once again, this shows that the authors' considerations about the distribution of WC particles across the coating (Fig.3 and in many places of the text) appear to contradict to their own experimental results.
Response:
We respectively disagree. In Fig7(b), the distribution of W and C elements are apparently not in agreement with the results discussed above. The additive atomic amount of WC is much less than that of HEA. Therefore, that may cause the inaccuracy of WC.
Caption to Fig.7, "mechanical properties of the HEA/WC coating", is misleading: cladding profile (a), element distribution curves (b) and polarization curves (d) have nothing to do with "mechanical properties".
Response:
We have changed drawing statement of Fig 7 from “the mechanical properties of the HEA/WC coating.” to “Fig. 7 (a) the cladding profile, (b) element distribution curves, (c) the microhardness distributions and (d) the potentiodynamic polarization evolution curves of coating”.
P.6, lines 179-180: According to the interval standard of 5-20% dilution rate, the upper dilution rate was reasonable." The phrase is confusing, the authors probably wanted to say that the obtained value is below the upper limit.
Response:
We feel sorry about our poor English. And the change to the sentence and reference at line 40 in the previous manuscript version is shown below: According to the interval standard of 5-20% dilution rate, the dilution rate of the cladded coating is reasonable.
P.7, lines 186-187: "... hard WC particles penetrated into the interface with Marangoni convection, the microhardness of this region increased gradually." First, the WC did not penetrate "into" the interface but are supposed to concentrate near it. Second, as I stated above, in the paper there are no evidences that the fraction of WC particles at the interface is higher than in the central part of the coating. Third, the Vickers hardness of WC is around 2600. If the WC particles really deposited at the interface, the microhardenss near the interface would be substantially higher than away from it. But in Fig.7(c) the situation is the opposite. So, once again, the authors' concept expressed in Fig.3 and a large part of the text concerning WC particles appears misleading.
Response:
Thanks for review’s professional suggestion. We have checked our data and logic multiply, and we accept review’s suggestion. The sentence in line 186-187 may cause mistakes and confused logic. So we delete the sentence of “In the HAZ, as hard WC particles penetrated into the interface with Marangoni convection, the microhardness of this region increased gradually.”
Conclusion (1): "The morphology ... was observed and explained by Marangoni effect and constitutional undercooling criterion." First, as shown above, the authors' explanation including the Marangoni effect is misleading. Second, the constitutional undercooling criterion is not used in this work.
Response:
Thanks for review’s professional suggestion. We have checked our data and logic multiply, and we accept review’s suggestion. The sentence in line 186-187 may cause mistakes and confused logic. So we delete the sentence of “ The morphology of HEA composite coating was observed and explained by Marangoni effect and constitutional undercooling criterion.”
Conclusion (2): "...the remaining diffraction peaks are identified as some metal carbide and WC." In describing the XDR results presented in Fig.5, the authors did not identify which particular "metal carbide" was it, but they mentioned complex carbide (M,W)C (see comments to Fig.5)
Response:
In effect, the XRD peak of "(M, W)C phase" can signify the mixture of MCs (metal carbides) and WC, MCs means FeC, CrC and AlC, which are consistent with the standard card of JCPDS: no. 35-0799, JCPDS: no.51-0997 and JCPDS: no. 35-0783 respectively.
Conclusion (3): "The main reasons are grain refinement strengthening and solid-solution hardening." In the text, the authors mentioned the presence of undissolved WC particles (p.7, line 198), but here they do not describe their contribution to hardening. Then why there was so many reasoning in the text about the role of WC particles?
Response:
Thanks for review’s professional suggestion. We checked our data and logic repeatedly, and we have tried to get the following points across. Firstly, the existence of WC particles are ascribed to incomplete melting during laser cladding. With the addition of WC, the micro-structure of claded coating changes obviously. Secondly, AlCoCrFeNi/WC coating have better microhardness than that of AlCoCrFeNi coating, because of the changed micro-structure of silver. Finally, AlCoCrFeNi/WC coating exhibits better better corrosion resistance due to reduced electron extraction efficiencies and/or parasitic light absorption.

Round 2
Reviewer 2 Report
Dear,
The author and co-authors have taken into account my comments. And thanks to the introduction of native English speakers in the review, the quality of the manuscript has improved.
I would recommend it for publication.
Best regards,
Reviewer
Author Response
Thank you for carefully reviewing our manuscript and accepting it. Here, once again, we express our heartfelt thanks.
Reviewer 3 Report
In the response to the reviewer's comment No.1 concerning the English language, the authors noted: "As English is not our mother tongue, it is not so easy to change the English of our manuscript properly."
Thus is a bad excuse. And in the revised version there are still many errors in the English language.
Then, naturally, a question arises: if the authors realize that, why did they submit their work to this journal, where the requirements to the quality of professional English are very high?
I can only recommend that the authors should completely rewrite the paper according to the comments below and then find a native English speaker who is familiar with this professional area and ask him/her to carefully read it.
---------------
Reviewer's comment to the text on p.4 in the 1st version:
"Besides, the authors did not explain what are the dark spots in Fig.4. Are they the WC particles, same as in Fig.3(d)? If so, then it is seen that their number is larger in the middle (Fig.4(c)) then in the bottom (Fig.4(d)) of the coating, which contradicts the author's concept presented in Fig.3(g)."
In the authors' response, they completely ignored this important issue and did not give an answer nor made any changes to the text.
---------------
Reviewer's comment to the 1st version:
P.5, lines 131-132: "the nucleation rate of the grain is much faster than the growth rate, and the columnar dendritic structure shown was formed..." From the crystallization theory it is known that dendrites grow in the direction opposite to the heat flow, and here the growth rate must be higher than the nucleation rate - otherwise the structure would be equiaxial.
Response:
Thanks for review’s professional suggestion. We accept the suggestion and change the sentence of “Therefore, the nucleation rate of the grain is much faster than the growth rate, and the columnar dendritic structure shown was formed” to “Therefore, the nucleation rate of the grain is much faster than the growth rate, and the columnar dendritic structure shown was formed in Fig. 4 (d)”
My response:
The modified text is erroneous. Obviously, the authors were not very attentive in reading the reviewer's comment, and they actually reproduced the original, physically incorrect phrase.
---------------
Reviewer's comments to the 1st version:
P.5, Fig.5: what is "(M, W)C phase" depicted in this figure? If it is a complex carbide, then which particular metal M (or metals) dissolves in WC? In the text (line 148) the authors speak only about WC and "some metal carbide", which actually was not identified whereas the carbides of the carbide-forming metals that are present in the HEA (Cr and Fe) are well known.
Response:
In effect, the XRD peak of "(M, W)C phase" can signify the mixture of MCs (metal carbides) and WC, MCs means FeC, CrC and AlC, which are consistent with the standard card of JCPDS: no. 35-0799, JCPDS: no.51-0997 and JCPDS: no. 35-0783 respectively.
The same answer is given by the authors to the reviewer's comment
My response:
This answer is erroneous. There aren't such carbides as FeC, CrC and AlC. The chromium carbides are Cr3C2, Cr7C3, and Cr23C6; the equilibrium iron carbide is Fe3C unless the authors mean metastable epsilon-carbide Fe2.4C; the aluminum carbide is Al4C3. So, the authors are supposed to make a new, more accurate identification of phases in their XRD spectrum or to perform a new XRD experiment.
-----------------
Reviewer's comment to the text on p.5 in the 1st version
"But then all the authors' consideration about the distribution of the WC particles (see Fig.3 and the comments above) appear incorrect. And what was the accuracy of EDS for C? It is known than for light elements it is very low."
Response: Thanks for your warm suggestion, we have changed Fig 6 instead of the old one.
My response:
The reviewer's comments concerning the distribution of the WC particles and the accuracy of EDS for C is not addressed in the modified version. Besides, I see no difference between the old and new versions of Fig.6.
-----------------
Reviewer's comment to the 1st version:
P.6, lines 179-180: According to the interval standard of 5-20% dilution rate, the upper dilution rate was reasonable." The phrase is confusing, the authors probably wanted to say that the obtained value is below the upper limit.
Response:
We feel sorry about our poor English. And the change to the sentence and reference at line 40 in the previous manuscript version is shown below: According to the interval standard of 5-20% dilution rate, the dilution rate of the cladded coating is reasonable.
My response:
Actually, as seen from the formula presented on p.10, the authors speak not about the "dilution rate" but about the dilution ratio; those are two different physical notions.
So, the English language and especially the usage of terminology should be carefully checked throughout the whole text.
-----------------
Reviewer's comment to the text on p.7 in the 1st version:
"Third, the Vickers hardness of WC is around 2600. If the WC particles really deposited at the interface, the microhardenss near the interface would be substantially higher than away from it. But in Fig.7(c) the situation is the opposite. So, once again, the authors' concept expressed in Fig.3 and a large part of the text concerning WC particles appears misleading."
Response:
Thanks for review’s professional suggestion. We have checked our data and logic multiply, and we accept review’s suggestion. The sentence in line 186-187 may cause mistakes and confused logic. So we delete the sentence of “In the HAZ, as hard WC particles penetrated into the interface with Marangoni convection, the microhardness of this region increased gradually.”
My response:
The correction made by the authors is insufficient. Same as the 1st version, the new version contradicts to the concept shown in Fig.3(e)-(g). Thus, the paper is still misleading.
-----------------
Reviewer's comment to the 1st version:
Conclusion (3): "The main reasons are grain refinement strengthening and solid-solution hardening." In the text, the authors mentioned the presence of undissolved WC particles (p.7, line 198), but here they do not describe their contribution to hardening. Then why there was so many reasoning in the text about the role of WC particles?
Response:
Thanks for review's professional suggestion. We checked our data and logic repeatedly, and we have tried to get the following points across. Firstly, the existence of WC particles are ascribed to incomplete melting during laser cladding. With the addition of WC, the micro-structure of claded coating changes obviously. Secondly, AlCoCrFeNi/WC coating have better microhardness than that of AlCoCrFeNi coating, because of the changed micro-structure of silver. Finally, AlCoCrFeNi/WC coating exhibits better corrosion resistance due to reduced electron extraction efficiencies and/or parasitic light absorption.
My response:
First, what is "changed micro-structure of silver"? There is no silver (i.e. Ag) in the material. Second, the text "better corrosion resistance due to reduced electron extraction efficiencies and/or parasitic light absorption" is confusing. The authors did not say a single word in the main text of the paper about "electron extraction" and about "parasitic light absorption". And how are the optical properties ("light absorption") connected to the corrosion resistance?
------------
In the review to the 1st version, I noted several times that the authors' concept shown in Fig.3 (e)-(g), where the WC particles segregate at the substrate/coating interface, is in contradiction to Fig.4 (WC particle distribution over the coating thickness) and Fig.7(c) (microhardness distribution across the coating).
The paper is still internally contradictive and, as outlined in my review to the 1st version, misleading. The authors made only minor changes, which can be regarded to as "cosmetic", and ignored many important comments concerning the links between different parts of the text. They did not pay due attention to results of XDR nor modified the concept of the paper. Probably, this is because of the fact that this journal gives a very short time for modifying the paper.
From the above it unambiguously follows that the revised version can not be published.
I find it necessary to suggest that the authors should reconsider the whole concept of the paper, perform more thorough analysis (in particular, XRD), rewrite it in a consistent manner paying special attention to eliminating internal contradictions between their statements concerning the structure formation, the distribution of WC particles and the internal links between the structure and properties. And only after that submit it to peer-reviewed journal, maybe a different one. Besides, thorough editing of the English text and terminology is recommended.
Author Response
In the response to the reviewer's comment No.1 concerning the English language, the authors noted: "As English is not our mother tongue, it is not so easy to change the English of our manuscript properly."
Thus is a bad excuse. And in the revised version there are still many errors in the English language.
Then, naturally, a question arises: if the authors realize that, why did they submit their work to this journal, where the requirements to the quality of professional English are very high?
I can only recommend that the authors should completely rewrite the paper according to the comments below and then find a native English speaker who is familiar with this professional area and ask him/her to carefully read it.
Response: Thank you for your suggestion. We tried our best to improve the English level, and checked our article for several times. We followed your suggestion and correct our errors. For example, our introduction is replaced as:
High entropy alloys (HEA) is composed of more than five principal components, which first reported by Yeh and Cantor et al in 2004 [1-2]. HEA have the same properties with traditional single principal component alloys, also, they have notable advantages such as outstanding thermal stability, fine abrasion, high strength, oxidation resistance and corrosion resistance [3-7]. Recently, HEA have been applied in many fields like aerospace, oil pipeline and automobile shell own to their Unique performance. Nowadays, the methods of induction melting and arc melting are widely used in the preparation of HEA. However, high cost of HEA limited their volume-produce and extensive use[8]. Up to now, researchers found that HEA can combine with steels as coating in the way of laser cladding, which can improve their mechanical properties. Further more, this method can also reduce the cost due to the high precision, low dissipation, and instantaneous heat proof capacity of laser cladding[9-10].
AlCoCrFeNi HEA coated materials have been reported in recent years, some of them obtained excellent properties such as high corrosion resistance, wear resistance and hardness [11-13]. More than that, the properties can be further improved through the method of adding ceramic powders in HEA coating as additive. For example, Li Yutao et al. synthesized in-situ TiC reinforced AlCoCrFeNi HEA coating by laser cladding method, and made an in-depth study on its wear resistance and hardness. They found that the AlCoCrFeNi-20% TiC coating had the best hardness and wear resistance, and the main wear mechanism of the HEA composite coating was abrasive wear[14]. Xiaofeng Li et al. prepared the AlCoCrFeNi-NbC HEA laser cladding coating, and found that NbC particles had a strong pinning effect, which inhibited the grain growth of the HEA coating[15]. By using high-velocity oxygen fuel spraying technology, Junpeng Lv et al. fabricated the AlCoCrFeNi-50%WC -10Co HEA composite coatings on 316L stainless steel (316Lss)[16]. At present, there are few investigations on AlCoCrFeNi HEA composite coatings with WC reinforcing particles on 316Lss by laser cladding.
In this paper, we use 10% mass fraction WC ceramic powder and AlCoCrFeNi HEA mixture as precursor of laser clad coating on the surface of 316Lss. We investigated micro-structure, phase composition, mirco-hardness and corrosion resistance of the compound coatings, and discussed the mechanism. The research can provide a reference for the application of AlCoCrFeNi HEA-WC composite coatings.
Reviewer's comment to the text on p.4 in the 1st version:
"Besides, the authors did not explain what are the dark spots in Fig.4. Are they the WC particles, same as in Fig.3(d)? If so, then it is seen that their number is larger in the middle (Fig.4(c)) then in the bottom (Fig.4(d)) of the coating, which contradicts the author's concept presented in Fig.3(g)."
In the authors' response, they completely ignored this important issue and did not give an answer nor made any changes to the text.
Response: Thanks for your suggestion, the dark particles in Fig. 4 are WC micro-particles. They are caused by the incomplete melting of WC. Fig 3(d) shows that WC particles appear at the bottom of cladded coating because of their high density. The WC particles in Fig 3 are millimetre-sized which attribute to metallographic microscope. However, size of WC micro-particles in Fig 4(c-d) are 2-4 μm,they can be shown because of high-resolution of electron microscope. The small size of WC micro-particles can break the limitation of its high density. This lead to the WC micro-particles can evenly distributed in cladded coating and larger WC particles can only appear at the bottom of cladded coating.
Reviewer's comment to the 1st version:
P.5, lines 131-132: "the nucleation rate of the grain is much faster than the growth rate, and the columnar dendritic structure shown was formed..." From the crystallization theory it is known that dendrites grow in the direction opposite to the heat flow, and here the growth rate must be higher than the nucleation rate - otherwise the structure would be equiaxial.
Response:
Thanks for reviewer’s professional suggestion. We accept the suggestion and change the sentence of “Therefore, the nucleation rate of the grain is much faster than the growth rate, and the columnar dendritic structure shown was formed” to “Therefore, the nucleation rate of the grain is much faster than the growth rate, and the columnar dendritic structure shown was formed in Fig. 4 (d)”
My response:
The modified text is erroneous. Obviously, the authors were not very attentive in reading the reviewer's comment, and they actually reproduced the original, physically incorrect phrase.
Response again: Thank you very much.
From the professional literature and knowledge accumulation that we are familiar with, our description and explanation should be reasonable. And references [22], [23] also have similar conclusions.
Reviewer's comments to the 1st version:
P.5, Fig.5: what is "(M, W)C phase" depicted in this figure? If it is a complex carbide, then which particular metal M (or metals) dissolves in WC? In the text (line 148) the authors speak only about WC and "some metal carbide", which actually was not identified whereas the carbides of the carbide-forming metals that are present in the HEA (Cr and Fe) are well known.
Response:
In effect, the XRD peak of "(M, W)C phase" can signify the mixture of MCs (metal carbides) and WC, MCs means FeC, CrC and AlC, which are consistent with the standard card of JCPDS: no. 35-0799, JCPDS: no.51-0997 and JCPDS: no. 35-0783 respectively.
The same answer is given by the authors to the reviewer's comment
My response:
This answer is erroneous. There aren't such carbides as FeC, CrC and AlC. The chromium carbides are Cr3C2, Cr7C3, and Cr23C6; the equilibrium iron carbide is Fe3C unless the authors mean metastable epsilon-carbide Fe2.4C; the aluminum carbide is Al4C3. So, the authors are supposed to make a new, more accurate identification of phases in their XRD spectrum or to perform a new XRD experiment.
Response again: Thanks for reviewer’s suggestion, we checked our XRD spectrum, and we modified our results. However, I can not totally agree with reviewer’s viewpoint. As well known, WC is melted partially during cladding process, then complex metal carbides are produced. For example, Anne Mertens’s group reported that the addition of WC in molten metal lead to a partial dissolution of the initial WC particles. And this method enriches the composition of substrate material. And (FeCr)23C6 and (FeW)6C are found in XRD patterns[17]. The XRD results have changed as follow:
Fig. 5 shows the XRD pattern of the AlCoCrFeNi/WC coating. It can be seen that the diffraction peaks with the highest intensity are identified as the solid-solution with a BCC phase as the major constituent, while the remaining diffraction peaks are identified as some metal carbide and WC which is similar to Penlin Zhang et al [24]. In effect, the XRD peak at 32o is ascribed to WC and peaks at about 43o and 45o can attribute to Fe3W3C[20]. Further more, peaks of (Cr,Fe)7C3 (JCPDS : no.05-0720) and Fe2W(JCPDS: no.03-0920) can be shown in Fig 5.
Reviewer's comment to the text on p.5 in the 1st version
"But then all the authors' consideration about the distribution of the WC particles (see Fig.3 and the comments above) appear incorrect. And what was the accuracy of EDS for C? It is known than for light elements it is very low."
Response: Thanks for your warm suggestion, we have changed Fig 6 instead of the old one.
My response:
The reviewer's comments concerning the distribution of the WC particles and the accuracy of EDS for C is not addressed in the modified version. Besides, I see no difference between the old and new versions of Fig.6.
Response: As reported, the addition of WC powders in molten metal lead to a partial or total dissolution of initial WC particles. As shown above, WC particles with different sizes consist in cladded coating, there are WC particles with large diameters deposit at the bottom of molten pool because of the high density, and WC micro-particles appear at middle and top of molten pool and are combined with HEA completely. Therefore, element of C can be found in hole molten pool.
Also, we changed Fig.6 to issue the definition.
Reviewer's comment to the 1st version:
P.6, lines 179-180: According to the interval standard of 5-20% dilution rate, the upper dilution rate was reasonable." The phrase is confusing, the authors probably wanted to say that the obtained value is below the upper limit.
Response:
We feel sorry about our poor English. And the change to the sentence and reference at line 40 in the previous manuscript version is shown below: According to the interval standard of 5-20% dilution rate, the dilution rate of the cladded coating is reasonable.
My response:
Actually, as seen from the formula presented on p.10, the authors speak not about the "dilution rate" but about the dilution ratio; those are two different physical notions.
So, the English language and especially the usage of terminology should be carefully checked throughout the whole text.
Response again: Thank you very much. We checked the mistakes and have already changed “dilution rate” to “dilution ratio”.
Reviewer's comment to the text on p.7 in the 1st version:
"Third, the Vickers hardness of WC is around 2600. If the WC particles really deposited at the interface, the microhardenss near the interface would be substantially higher than away from it. But in Fig.7(c) the situation is the opposite. So, once again, the authors' concept expressed in Fig.3 and a large part of the text concerning WC particles appears misleading."
Response:
Thanks for review’s professional suggestion. We have checked our data and logic multiply, and we accept review’s suggestion. The sentence in line 186-187 may cause mistakes and confused logic. So we delete the sentence of “In the HAZ, as hard WC particles penetrated into the interface with Marangoni convection, the microhardness of this region increased gradually.”
My response:
The correction made by the authors is insufficient. Same as the 1st version, the new version contradicts to the concept shown in Fig.3(e)-(g). Thus, the paper is still misleading.
Response: Thanks for your suggestion, the dark particles in Fig. 4 are WC micro-particles. They are caused by the incomplete melting of WC. Fig 3(d) shows that WC particles appear at the bottom of cladded coating because of their high density. The WC particles in Fig 3 are millimetre-sized which attribute to metallographic microscope. However, size of WC micro-particles in Fig 4(c-d) are 2-4 μm,they can be shown because of high-resolution of electron microscope. The small size of WC micro-particles can break the limitation of its high density. This lead to the WC micro-particles can evenly distributed in cladded coating and larger WC particles can only appear at the bottom of cladded coating.
Reviewer's comment to the 1st version:
Conclusion (3): "The main reasons are grain refinement strengthening and solid-solution hardening." In the text, the authors mentioned the presence of undissolved WC particles (p.7, line 198), but here they do not describe their contribution to hardening. Then why there was so many reasoning in the text about the role of WC particles?
Response:
Thanks for review's professional suggestion. We checked our data and logic repeatedly, and we have tried to get the following points across. Firstly, the existence of WC particles are ascribed to incomplete melting during laser cladding. With the addition of WC, the micro-structure of claded coating changes obviously. Secondly, AlCoCrFeNi/WC coating have better microhardness than that of AlCoCrFeNi coating, because of the changed micro-structure of silver. Finally, AlCoCrFeNi/WC coating exhibits better corrosion resistance due to reduced electron extraction efficiencies and/or parasitic light absorption.
My response:
First, what is "changed micro-structure of silver"? There is no silver (i.e. Ag) in the material. Second, the text "better corrosion resistance due to reduced electron extraction efficiencies and/or parasitic light absorption" is confusing. The authors did not say a single word in the main text of the paper about "electron extraction" and about "parasitic light absorption". And how are the optical properties ("light absorption") connected to the corrosion resistance?
Response: I am sorry for our mistakes, these words come from a unmodified version. And we have already changed the errors.
In the review to the 1st version, I noted several times that the authors' concept shown in Fig.3 (e)-(g), where the WC particles segregate at the substrate/coating interface, is in contradiction to Fig.4 (WC particle distribution over the coating thickness) and Fig.7(c) (microhardness distribution across the coating).
The paper is still internally contradictive and, as outlined in my review to the 1st version, misleading. The authors made only minor changes, which can be regarded to as "cosmetic", and ignored many important comments concerning the links between different parts of the text. They did not pay due attention to results of XDR nor modified the concept of the paper. Probably, this is because of the fact that this journal gives a very short time for modifying the paper.
From the above it unambiguously follows that the revised version can not be published.
I find it necessary to suggest that the authors should reconsider the whole concept of the paper, perform more thorough analysis (in particular, XRD), rewrite it in a consistent manner paying special attention to eliminating internal contradictions between their statements concerning the structure formation, the distribution of WC particles and the internal links between the structure and properties. And only after that submit it to peer-reviewed journal, maybe a different one. Besides, thorough editing of the English text and terminology is recommended.
Response: We are pleasure to take reviewer’s pertinent suggestions and these are beneficial to our process. We have checked our errors carefully and did our best to amend them. We did heavy workload in utmost sincerity, and we request editor consider our manuscript carefully.
For example: the revised introduction is shown as follow:
High entropy alloys (HEA) is composed of more than five principal components, which first reported by Yeh and Cantor et al in 2004 [1-2]. HEA have the same properties with traditional single principal component alloys, also, they have notable advantages such as outstanding thermal stability, fine abrasion, high strength, oxidation resistance and corrosion resistance [3-7]. Recently, HEA have been applied in many fields like aerospace, oil pipeline and automobile shell own to their Unique performance. Nowadays, the methods of induction melting and arc melting are widely used in the preparation of HEA. However, high cost of HEA limited their volume-produce and extensive use[8]. Up to now, researchers found that HEA can combine with steels as coating in the way of laser cladding, which can improve their mechanical properties. Further more, this method can also reduce the cost due to the high precision, low dissipation, and instantaneous heat proof capacity of laser cladding[9-10].
AlCoCrFeNi HEA coated materials have been reported in recent years, some of them obtained excellent properties such as high corrosion resistance, wear resistance and hardness [11-13]. More than that, the properties can be further improved through the method of adding ceramic powders in HEA coating as additive. For example, Li Yutao et al. synthesized in-situ TiC reinforced AlCoCrFeNi HEA coating by laser cladding method, and made an in-depth study on its wear resistance and hardness. They found that the AlCoCrFeNi-20% TiC coating had the best hardness and wear resistance, and the main wear mechanism of the HEA composite coating was abrasive wear[14]. Xiaofeng Li et al. prepared the AlCoCrFeNi-NbC HEA laser cladding coating, and found that NbC particles had a strong pinning effect, which inhibited the grain growth of the HEA coating[15]. By using high-velocity oxygen fuel spraying technology, Junpeng Lv et al. fabricated the AlCoCrFeNi-50%WC -10Co HEA composite coatings on 316L stainless steel (316Lss)[16]. At present, there are few investigations on AlCoCrFeNi HEA composite coatings with WC reinforcing particles on 316Lss by laser cladding.
In this paper, we use 10% mass fraction WC ceramic powder and AlCoCrFeNi HEA mixture as precursor of laser clad coating on the surface of 316Lss. We investigated micro-structure, phase composition, mirco-hardness and corrosion resistance of the compound coatings, and discussed the mechanism. The research can provide a reference for the application of AlCoCrFeNi HEA-WC composite coatings.

Round 3
Reviewer 3 Report
The manuscript has been improved, some contradictions are removed, but not all.
But there are still a lot of grammar, stylistic and terminological errors in the English language.
Some examples (but not all), are presented below. It should be noted that it is not the reviewers’ duty to pint out to all of the language errors.
P.2: "Recently, HEA have been applied in many fields like aerospace, oil pipeline, and automobile shell, and so on own to their Unique performance." What is "automobile shell"? And why "own"?
P.2: "volume-produce" – what is this? Why not mass production?
P.3, Section 2 Experimental methods: "The average diameter of HEA powder is 70um, while the average diameter of WC particle is 45um.” What is um? Do the authors mean mm (micron)?
P.6: "...smaller particles may be distributed in hole coating..." What is a " hole coating"?
P.6: "... predecessor powders ... are promixed for over 2 hours, in order to issue the uniformity of the sample": Incorrect phrase from the viewpoint of both terminology, spelling and style; term "predecessor" is used only in History.
P.6: "... temperature of coating enhanced...". It’s an error in terminology: how can temperature enhance?
P.6: "... distributed in hole coating." What is a "hole coating"?
P.6-7: "... there are few thermal crackings and low porosity points." Cracking is the process of crack formation or a petrochemical term. And what are the "porosity points"?
P.9: " The EDS mappings can perform more intuitive understanding the distribution of elements...” The text contains grammar and stylistic errors.
And there are some other errors.
Thus, it is strongly recommended that the authors have the manuscript be carefully read by a native English speaker who is familiar with Materials Science.
There are still certain contradictions in the text. On p.6 it is stated: "The larger WC particles can deposit at the bottom of the coating". This is shown schematically (i.e. theoretically) in Fig.3(g).
Later on P.7 it is written: "The WC particles in Fig 3 are millimetre-sized which attribute to metallographic microscope. However, size of WC micro-particles in Fig 4(c-d) are 2-4 μm,they can be shown because of high-resolution of electron microscope. ... This lead to the WC micro-particles can evenly distributed in cladded coating and WC particles can only appear at the bottom of cladded coating."
First, if the white circles in Fig.3(b-d) are large WC particles, then their size is about 50-100 micron, so they are not "millimetre-sized". Second, in Fig 4(d) the scale bar is 20 micron, so the whole width of this figure is about 200 micron.
Hence, any reader/reviewer will immediately ask the following questions:
(1) Where are the larger, "millimetre-sized" WC particles in Fig.4(d), which, as stated in caption to Fig.4, shows the "bottom regions" of the coating?
(2) Why the authors have not chosen another area in the "bottom region" where the large WC particles are present, in order to prove experimentally their concept presented in Fig.3 and the relevant text? May be because there are no large particles, and hence the concept is incorrect?
Thus, on this stage manor revision is recommended (corrections to methodological errors and text editing).
Author Response
Dear editor and reviewers:
Thank you for reviewing our manuscript again. The reviewer for his/her efforts in reviewing our manuscript, overall positive assessment and providing constructive suggestions. We have revised the manuscript carefully, which are addressed point-by-point below:
For reviewer 3:
The manuscript has been improved, some contradictions are removed, but not all. But there are still a lot of grammar, stylistic and terminological errors in the English language.
Some examples (but not all), are presented below. It should be noted that it is not the reviewers’ duty to pint out to all of the language errors.
Response: Thanks for your affirmation of the execution and results for our work. We are so grateful for your instructive advice, which can enhance our English level. We have checked the manuscript carefully and corrected the mistakes as following:
P.2: "Recently, HEA have been applied in many fields like aerospace, oil pipeline, and automobile shell, and so on own to their Unique performance." What is "automobile shell"? And why "own"?
Response: I am sorry for our careless, the sentence have been changed to “Recently, HEA have been applied in many fields like aerospace, oil pipeline, automobile shell because of their unique performance”
P.2: "volume-produce" – what is this? Why not mass production?
Response: Your suggestion is useful, and we have changed the sentence as “However, high cost of HEA limited their mass production and extensive use”
P.3, Section 2 Experimental methods: "The average diameter of HEA powder is 70 um, while the average diameter of WC particle is 45 um.” What is um? Do the authors mean mm (micron)?
Response: We are sorry for our careless, the “um” have been changed to “μm”.
P.6: "...smaller particles may be distributed in hole coating..." What is a " hole coating"?
Response: We are sorry of making an stupid mistake, and we have changed “hole” to “whole”.
P.6: "... predecessor powders ... are promixed for over 2 hours, in order to issue the uniformity of the sample": Incorrect phrase from the viewpoint of both terminology, spelling and style; term "predecessor" is used only in History.
Response: Thanks for your correction and we have already changed "predecessor" to “precursor”.
P.6: "... temperature of coating enhanced...". It’s an error in terminology: how can temperature enhance?
Response: Thanks for your correction, we have changed “During laser cladding, the temperature of coating enhanced quickly due to the laser beams energy” to “The temperature of the coating increased quickly due to the high laser beams energy during cladding”
P.6: "... distributed in hole coating." What is a "hole coating"?
Response: We are sorry of making an stupid mistake, and we have changed “hole” to “whole”.
P.6-7: "... there are few thermal crackings and low porosity points." Cracking is the process of crack formation or a petrochemical term. And what are the "porosity points"?
Response: Thanks for the suggestion, and we have changed the sentence to “there are few fractures ”
P.9: " The EDS mappings can perform more intuitive understanding the distribution of elements...” The text contains grammar and stylistic errors.
Response: We have checked the grammar and changed the sentence from “The EDS mappings can perform more intuitive understanding the distribution of elements around the interface of the cladded coatings, as shown in Fig. 6” to “Fig 6 shows the EDS mappings of the coating with different elements”
And there are some other errors.
Thus, it is strongly recommended that the authors have the manuscript be carefully read by a native English speaker who is familiar with Materials Science.
Response: Thanks for your suggestion, we have invited a professional author to correct our manuscript.
There are still certain contradictions in the text. On p.6 it is stated: "The larger WC particles can deposit at the bottom of the coating". This is shown schematically (i.e. theoretically) in Fig.3(g).
Later on P.7 it is written: "The WC particles in Fig 3 are millimetre-sized which attribute to metallographic microscope. However, size of WC micro-particles in Fig 4(c-d) are 2-4 μm,they can be shown because of high-resolution of electron microscope. ... This lead to the WC micro-particles can evenly distributed in cladded coating and WC particles can only appear at the bottom of cladded coating."
First, if the white circles in Fig.3(b-d) are large WC particles, then their size is about 50-100 micron, so they are not "millimetre-sized". Second, in Fig 4(d) the scale bar is 20 micron, so the whole width of this figure is about 200 micron.
Hence, any reader/reviewer will immediately ask the following questions:
- Where are the larger, "millimetre-sized" WC particles in Fig.4(d), which, as stated in caption to Fig.4, shows the "bottom regions" of the coating?
Response: The Fig 4 show the low and high resolution SEM images of WC-added HEA coating. Fig. 4 (a) shows the section of the laser cladded coating, several WC particles can be seen at the bottom of the coating with diameters of 30-50 μm, as described above. Fig. 4 (b)-(d) exhibit the high-resolution SEM images of the top, middle and bottom areas, respectively. The SEM images of Fig. 4(b)-(d) mainly shows the difference crystal structures in different areas. Therefore, we choose the area of bottom coating without WC particles, to insure the integrity and sharpness of SEM image, and the crystal structures can be seen clearly in Fig 4(d).
- Why the authors have not chosen another area in the "bottom region" where the large WC particles are present, in order to prove experimentally their concept presented in Fig.3 and the relevant text? May be because there are no large particles, and hence the concept is incorrect?
Response: In effect, Fig 3(a)-(d) and Fig 4 characteristic the same sample. In order to ensure the crystal structure at the bottom of the coating, both sharpness and uniformity are important indications. If we choose the high-resolution SEM image with WC particles, we can not give consideration to both uniformity and sharpness.
Thus, on this stage manor revision is recommended (corrections to methodological errors and text editing).
